# Arsenicals, the Integrated Stress Response, and Epstein–Barr Virus Lytic Gene Expression

**DOI:** 10.3390/v13050812

**Published:** 2021-04-30

**Authors:** Jaeyeun Lee, Jennifer Stone, Prashant Desai, John G. Kosowicz, Jun O. Liu, Richard F. Ambinder

**Affiliations:** 1Departments of Oncology, Johns Hopkins University School of Medicine, Baltimore, MD 21287, USA; jannjaeyeun@gmail.com (J.L.); jstone29@jhmi.edu (J.S.); pdesai1@jhmi.edu (P.D.); john.kosowicz@gmail.com (J.G.K.); joliu@jhu.edu (J.O.L.); 2Departments of Pharmacology and Molecular Sciences, Johns Hopkins University School of Medicine, Baltimore, MD 21287, USA

**Keywords:** Epstein–Barr virus, integrated stress response, arsenic

## Abstract

Following our observation that clofoctol led to Epstein–Barr virus (EBV) lytic gene expression upon activation of the integrated stress response (ISR), we decided to investigate the impact of As_2_O_3_ on viral lytic gene expression. As_2_O_3_ has also been reported to activate the ISR pathway by its activation of the heme-regulated inhibitor (HRI). Our investigations show that As_2_O_3_ treatment leads to eIF2α phosphorylation, upregulation of ATF4 and TRB3 expression, and an increase of EBV Zta gene expression in lymphoid tumor cell lines as well as in naturally infected epithelial cancer cell lines. However, late lytic gene expression and virion production were blocked after arsenic treatment. In comparison, a small molecule HRI activator also led to increased Zta expression but did not block late lytic gene expression, suggesting that As_2_O_3_ effects on EBV gene expression are also mediated through other pathways.

## 1. Introduction

The Epstein–Barr virus (EBV), a human gammaherpesvirus, is associated with a variety of malignancies including some lymphomas of B, T, and NK cell origin, and carcinomas including nasopharyngeal and gastric carcinomas [1,2]. In tumors, latency gene expression predominates and there is little or no expression of the viral proteins associated with virion production. While some have argued that lytic gene expression plays an important role in the transformation and maintenance of malignancy, others have argued that pharmacologic induction of viral lytic gene expression might be a therapeutic strategy to treat these cancers [3,4,5]. EBV reactivation may also be important in a variety of chronic diseases [6].

Arsenic trioxide (As_2_O_3_) has been reported to activate EBV lytic gene expression in epithelial cells [7,8] but not in Burkitt lymphoma cells [9]. As_2_O_3_ is used in the clinic in the treatment of promyelocytic leukemia [10]. In that disease which is characterized by a fusion of the PML and RARα genes, As_2_O_3_ reacts with the RING-finger domain of PML leading to SUMOylation and degradation of the fusion protein [11]. 

In recent work, we reported that eIF2α phosphorylation, by the protein kinase R (PKR)-like endoplasmic reticulum (ER) kinase (PERK), activated EBV lytic viral transcription [12]. Phosphorylation of eIF2α has been extensively studied in other contexts [13]. A variety of cellular stresses lead to its phosphorylation at Ser51, stalling of translation initiation complexes, and inhibition of global protein synthesis. This pathway is known as the integrated stress response (ISR). Four kinases phosphorylate this eIF2α serine: PERK, heme-regulated inhibitor kinase (HRI), general control nonderepressible 2 (GCN2), and double-stranded RNA-dependent protein kinase (PKR). Homologous in their catalytic domains, the regulatory domains of these kinases respond to different stresses. PERK is activated by endoplasmic reticulum stress associated with the accumulation of unfolded proteins. HRI is expressed at the highest levels in erythroid cells where it is activated by a deficiency in heme. It regulates globin mRNA translation as a function of the availability of heme. PKR is induced by class 1 interferons and activated by binding to highly structured double-stranded RNAs. GCN2 is activated by binding uncharged tRNAs that are associated with amino acid starvation. With the observation that PERK-mediated phosphorylation of eIF2α led to activation of EBV lytic gene expression [12], we were interested in the possibility that the other arms of the ISR might also mediate activation of lytic gene expression, including HRI, which is activated by As_2_O_3_ (Figure 1A) [14,15,16]. 

In the experiments reported here, we sought to better understand the impact of arsenicals and other ISR activators on EBV lytic gene expression.

## 2. Materials and Methods

### 2.1. Cell Culture

BX1-Akata, an engineered derivative of the Akata EBV (+) Burkitt lymphoma cell line, which carries a recombinant EBV expressing GFP, was a gift from L. Hutt-Fletcher (Louisiana State University) [17]. Raji is an EBV (+) Burkitt lymphoma cell line. SNU719 is a naturally derived EBV (+) gastric cancer cell line and was a gift from J.M. Lee (Yonsei University) [18]; C666-1 is an EBV (+) nasopharyngeal carcinoma cell line and was a gift from S. Diane Hayward [19]. LCL is an EBV-immortalized lymphoblastoid cell line established in the lab by infecting normal B lymphocytes with EBV strain B95-8. All cell lines were cultured in RPMI 1640, 2 mM L-glutamine, 100 μg/mL streptomycin, 100 IU/mL penicillin, and 10% v/v fetal bovine serum (FBS). Additionally, 500 μg/mL G418 (Geneticin; Gibco™ by Life Technologies, New York, NY, USA) was added for the BX1-Akata cell line.

### 2.2. Reagents

Arsenic trioxide, sodium arsenite, BTdCPU, and anti-IgG were purchased from MilliporeSigma (Burlington, MA, USA). 

### 2.3. qRT-PCR

RNeasy Mini Kit (QIAGEN, Germantown, MD, USA) was used for RNA extraction, and iScript reverse synthase kit (Bio-Rad, Hercules, CA, USA) was used to reverse-transcribe the RNA into cDNA. SsoFast Evagreen Supermix (Bio-Rad, Hercules, CA, USA) with 500 nM primers and cDNA corresponding to 20 ng of the RNA were used for each reaction of qPCR. cDNA was amplified at 95 °C for 30 s for 1 cycle and 95 °C for 5 s and 60 °C for 10 s, for a total of 40 cycles in a CFX96 real-time thermocycler. EBV primers used were Zta Forward (5′-ACATCTGCTTCAACAGGAGG-3′), Zta Reverse (5′-AGCAGACATTGGTGTTCCAC-3′), BMRF1 Forward (5′-CTAGCCGTCCTGTCCAAGTGC-3′), BMRF1 Reverse (5′-AGCCAAACGCTCCTTGCCCA-3′), gp350 Forward (5′-GTCAGTACACCATCCAGAGCC-3′), gp350 Reverse (5′-TTGGTAGACAGCCTTCGTATG-3′), gp110 Forward (5′-AACCTTTGACTCGACCATCG-3′), and gp110 Reverse (5′-ACCTGCTCTTCGATGCACTT-3′). Trib3 primers were Forward (5′-CGTGATCTCAAGCTGTGTCG-3′) and Trib3 Reverse (5′-AGCTTCTTCCTCTCACGGTC-3′). GAPDH primers Forward (5′-TCTTTTGCGTCGCCAGCCGA-3′) and GAPDH Reverse (5′-AGTTAAAAGCAGCCCTGGTGACCA-3′) were used as a control. The HRI primer set was purchased from Santa Cruz Biotechnology. The gene expression was normalized to GAPDH expression by using the comparative Ct method and presented as the fold change relative to the untreated control. 

### 2.4. Immunoblots

For protein extractions, 1 × 10^7^ cells were washed in PBS, and the pellets were resuspended in RIPA buffer containing 1× protease/phosphatase inhibitor cocktail (Santa Cruz Biotechnology, Dallas, TX, USA). After 15 min incubation in ice and 15 min of rotation at 4 °C, the proteins were isolated by centrifugation at 13,000 rpm at 4 °C for 5 min and collected from the supernatant. Equal amounts of protein (30 µg) per sample were separated by SDS-PAGE and subsequently transferred to nitrocellulose membranes. Western blotting were performed with antibodies against EBV Zta, BMRF1, ATF4 (Santa Cruz Biotechnology, Dallas, TX, USA), p-eIF2α (Abcam, Cambridge, UK), PML (Novus Biologicals, Littleton, CO, USA), Actinin (Cell Signaling technology, Danvers, MA, USA), and β-actin (Sigma-Aldrich, St. Louis, MO, USA). ECL chemiluminescent detection reagents (GE Healthcare, Chicago, IL, USA) with autoradiography film (Thomas Scientific, Swedesboro, NJ, USA) were used for the detection.

### 2.5. Immunofluorescence 

Next, 1.5 × 10^5^ cells were spun onto microscope slides using a Cytospin centrifuge and fixed using ice-cold methanol for 15 min. After blocking with 5% Bovine serum albumin (BSA) in PBS for 30 min, cells were stained with anti-EBV Zta or gp350 antibody (Santa Cruz, Dallas, TX, USA) for 1 h, washed three times for ten minutes each with 5% BSA, 0.1% Tween-20 in PBS, and incubated with Cy3 goat anti-mouse antibody (Jackson Immunoresearch, West Grove, PA, USA) for 1 h at room temperature. After three final washes, cells were stained with Vectashield mounting media with DAPI (Vector Laboratories, Burlingame, CA, USA), and a ZOE Fluorescent cell imager (Bio-Rad) was used for the fluorescence detection.

### 2.6. shRNA Knockdown

A pool of lentiviral particles containing 3 different shRNA constructs targeting HRI, a shRNA targeting PML, and control lentiviral particles encoding a scrambled sequence (Santa Cruz Biotechnology, Dallas, TX, USA) were used according to the manufacturer’s protocol, and the stable cell lines expressing the shRNA were selected with puromycin. 

### 2.7. Raji Infection Assay

BX1-Akata cells (1.5 × 10^7^) were treated with anti-IgG (10 μg/mL), arsenic trioxide (10 µM), or sodium arsenite (10 µM) and incubated for 4 days. After spinning the cells, the supernatant was passed through a Millex-HV Syringe Filter Unit (0.45 µm, Milli-poreSigma, Burlington, MA, USA), concentrated with a centrifugal filter (Amicon Ultra-15 Centrifugal Filter Unit, MilliporeSigma, Burlington, MA, USA), and 0.2 mL was used to infect Raji cells (2 × 10^5^ in 1 mL medium). TPA (20 ng/mL) and NaB (3 mM) were added 24 h after the infection and the GFP-positive cells were counted the next day using a ZOE Fluorescent cell imager.

## 3. Results

### 3.1. Arsenical-Induction of the ISR

We first assessed markers of ISR induction in an EBV (+) Burkitt lymphoma cell line, BX1-Akata, following treatment with As_2_O_3_. Phosphorylation of eIF2α and increased ATF4 protein accumulation were observed after this treatment (Figure 1B). Parallel results were seen with *N*,*N*′-diarylurea, BTdCPU (1-(benzo[*d*][1,2,3]thiadiazol-6-yl)-3-(3,4-dichlorophenyl)urea). This small molecule has been previously identified as directly interacting with HRI and inducing eIF2α phosphorylation [15]. As_2_O_3_ treatment also led to a time- and dose-dependent increase in Trib3 RNA (also known as NIPK and SKIP3) (Appendix A). These results are consistent with As_2_O_3_-mediating ISR activation [20,21,22]. We also observed that As_2_O_3_ treatment leads to degradation of PML, whereas BTdCPU had no effect on PML expression as judged by immunoblot methods (Figure 1B).

### 3.2. EBV Immediate Early Lytic Gene Expression Is also Activated

BX1-Akata cells were treated with varying doses of As_2_O_3_ or sodium arsenite (NaAsO_2_), and EBV Zta RNA levels were examined by qRT-PCR. Zta expression increased in a dose-dependent and time-dependent manner at concentrations of As_2_O_3_ that are achieved clinically in the treatment of promyelocytic leukemia (2–5 µM) [23] (Figure 2A). Increased Zta expression was also confirmed by immunofluorescence (Figure 2B). We note that the percentage of cells expressing Zta is high and exceeds activation typically seen in comparable experiments with TPA and sodium butyrate. The BX1-Akata cell line carries a recombinant viral genome that expresses green fluorescent protein (GFP) under control of the cytomegalovirus promoter. In the absence of lytic induction, only rare cells are lytic and only rare cells express GFP. With lytic induction, the numbers of cells expressing GFP also increase (Figure 2C). To investigate the relationship of the HRI pathway to activation of Zta expression by arsenicals, we performed a genetic knockdown experiment with a pool of three HRI shRNA constructs. HRI shRNA inhibited HRI RNA expression and substantially blocked arsenical induction of Trib3 and Zta (Figure 3A). Induction of lytic replication as judged by GFP expression was also blocked (Figure 3B). These results are consistent with the interpretation that the effects of arsenicals on Trib3 and Zta expression are mediated by HRI activation of the ISR. When HRI expression is blocked by shRNA constructs, the lytic response to the arsenicals is blocked.

### 3.3. Lytic Induction in Epithelial Cancer Cell Lines and LCLs

To better assess the generality of the findings with the Akata Burkitt line, we investigated the impact of arsenicals on naturally EBV-infected gastric carcinoma (SNU719) and nasopharyngeal carcinoma (C666-1) cell lines. Similar to the results seen in the Burkitt cell line, Trib3 RNA and Zta RNA increased following As_2_O_3_ treatment (Figure 4A,B). However, by immunofluorescence Zta protein was only marginally increased in SNU719 cells, and no increase could be detected in C666-1 cells). Zta RNA expression in an EBV lymphoblastoid cell line was activated by arsenic compounds (Figure 4C), but increased Zta protein expression was not observed following arsenical treatment). Thus, we conclude that the effects of arsenicals on Zta RNA were not limited to lymphoma cells—but note that protein levels varied substantially among cell lines. 

### 3.4. Arsenicals Have Different Impacts on Expression of a Delayed Early Gene and a Late Gene 

Although EBV Zta RNA and protein expression were induced by arsenic treatment, the effects on BMRF1, a delayed early gene, were quite different. RNA levels increased (Figure 5A) whereas protein levels decreased (Figure 5B). RNA and protein levels (as judged by immunofluorescence) of the late lytic gene gp350 did not increase following arsenic treatment (Figure 5C,D). To further evaluate the impact of arsenicals on lytic infection, we assessed virion production. For this purpose, we relied on the Raji infection assay [24,25]. In this assay, virions from BX1-Akata cells that infect Raji cells lead to the induction of the GFP expression in Raji cells. Lytic induction of BX1-Akata cells with anti-IgG yielded a supernatant that induced GFP expression in Raji cells and served as a positive control as can be seen in Figure 5E. In contrast, the supernatant from arsenical-treated BX1-Akata cells yielded no GFP signal. Thus following arsenic treatment, the immediate early Zta RNA and protein were expressed and the delayed early BMRF1 RNA was induced, but the BMRF1 protein was not detected. The late gp350 RNA was not induced and the protein was not detected and was consistent with a block in delayed early and late gene expression; virion production was inhibited (relative to baseline).

### 3.5. Effects of Direct HRI Activator on EBV Lytic Gene Expression

With the unexpected discordance between the impact of arsenicals on immediate early vs. delayed early and late genes, we were interested in investigating the impact of the small-molecule HRI activator BTdCPU on viral gene expression. As already noted (Figure 1B), BTdCPU led to eIF2α phosphorylation and ATF4 expression, but in contrast to the arsenicals, BTdCPU did not impact PML expression. With regard to viral gene expression, BTdCPU resulted in increased GFP expression in BX1-Akata cells (Figure 6A). When BX1-Akata cells were treated with As_2_O_3_, BTdCPU, or the combination, both drugs increased Trib3 RNA. The combination of As_2_O_3_ and BTdCPU markedly increased Zta RNA. In contrast, while both drugs individually increased BMRF1 RNA, the combination didn’t lead to further increase. For the late lytic genes, gp110 and gp350, As_2_O_3_ did not lead to an increase in RNA, but BTdCPU did. Used in combination, As_2_O_3_ inhibited BTdCPU-induction of late lytic gene expression (Figure 6B).

The comparison and the use of the two agents in combination made clear that the effects of As_2_O_3_ on late lytic gene expression were not strictly limited to effects mediated by eIF2α phosphorylation. One explanation might be that degradation of PML protein following As_2_O_3_ might account for the difference. In order to investigate whether the impact of As_2_O_3_ could be mimicked by BTdCPU in combination with a PML knockdown, we used a target-specific PML shRNA construct (Figure 6C). As shown in Figure 6D, knockdown of PML expression was not sufficient to block gp110 and gp350 RNA expression induced by BTdCPU treatment. Thus, it would appear that As_2_O_3_ effects cannot be entirely explained by effects on HRI or PML.

## 4. Discussion

These investigations confirm that HRI activation of the ISR is associated with upregulation of immediate early, early, and late EBV RNA expression and that As_2_O_3_ activates the ISR pathway. However, As_2_O_3_ activation of EBV genes is complex. While stimulating Zta and BMRF1 RNA expression as well as Zta protein expression, As_2_O_3_ inhibits BMRF1 protein expression and does not lead to an increase in late gp110 or gp350 RNA expression.

We have used GFP expression by the BX1 cell line as an indicator of lytic gene expression in this and several previous investigations [12,26]. GFP expression is appreciated in only a small percentage of cells under basal conditions. This percentage increases with lytic activation. Thus, as in this report, we typically see Zta protein expression in a larger percentage of cells than we see GFP expression. Similarly, others have used BX1 virus to superinfect the Raji EBV BL cell line and only appreciate GFP signal after lytic induction with TPA [27]. GFP expression in these cell lines is not marking the presence of the viral genome but lytic induction. 

Previous investigators have studied the impact of arsenicals on EBV gene expression [7,8,9,28]. Sides et al. reported activation of Zta and BMRF1 RNA expression in epithelial cells [8]. Yin et al. studied EBV-positive lymphoma cell lines treated with As_2_O_3_ at nanomolar concentrations for three days or longer. They reported decreased cell viability and reduced expression of Zta and BMRF1 [9]. Our results parallel those in the previous reports in some regards but differ in others, possibly reflecting differences in particular cell lines studied or in the duration of drug treatment. Our investigations also differ insofar as we report on the ISR pathway and HRI and studied BTdCPU, a direct activator of HRI.

Study of As_2_O_3_ and BTdCPU made clear that some effects of As_2_O_3_ are easily explained by the effect on HRI, while other effects must be mediated by other pathways. As_2_O_3_ effects on PML protein and fusion proteins have been investigated in great detail, as these effects underlie its therapeutic effect in the promyelocytic leukemia [11,29,30]. PML shRNA knockdown in combination with BTdCPU did not replicate the effects of As_2_O_3_ on late viral genes, so other pathways are likely involved. We note that As_2_O_3_ has been shown to induce oxidative stress, DNA damage, and mitochondrial stress, and these pathways may be important for the effects on late viral gene expression [31,32]. We should note that several EBV lytic genes (BZLF1, BRLF1, BGLF4) disrupt PML nuclear bodies through PML dispersal [33,34].

As has been true with other lytic activators, we found that activation of Burkitt cell lines was stronger than naturally infected epithelial cell lines [12,35]. However, although the extent of activation varied, activation of Zta was seen in all cell lines studied.

Differential effects of As_2_O_3_ and NaAsO_2_ at increasing BZLF1 transcription in a lymphoblastoid cell line were previously reported [28]. However, our findings did not show any differences between As_2_O_3_ and NaAsO_2_.

The effects of As_2_O_3_ we report are at levels achieved in leukemia patients with As_2_O_3_ [23]. The abortive lytic infection is of interest because activation of immediate early proteins may allow EBV-specific T cells (either a patient’s own or the result of adoptive immunotherapy) to more readily target tumor cells. At the same time, the blockade of delayed early and late lytic protein expression may protect against unwanted effects of lytic activation that some have hypothesized might help drive tumorigenesis or have other adverse effects. 

## Figures and Tables

**Figure 1 viruses-13-00812-f001:**
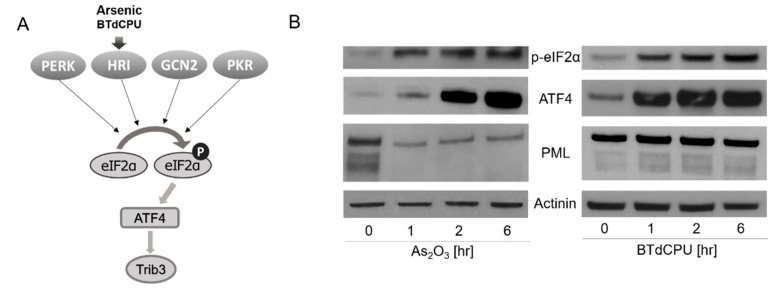
Arsenic and BTdCPU activate the p-eIF2α-ATF4 pathway in a lymphoma cell line. (**A**) A diagram illustrating how arsenic and BTdCPU activate the p-eIF2α-ATF4 pathway. (**B**) BX1-Akata cells were treated with 10 µM As_2_O_3_ or BTdCPU for the indicated time period, and expression of p-eIF2α, ATF4, and PML protein accumulation were determined by western blot. The cell lysate proteins on the blot were also incubated with actin antibodies.

**Figure 2 viruses-13-00812-f002:**
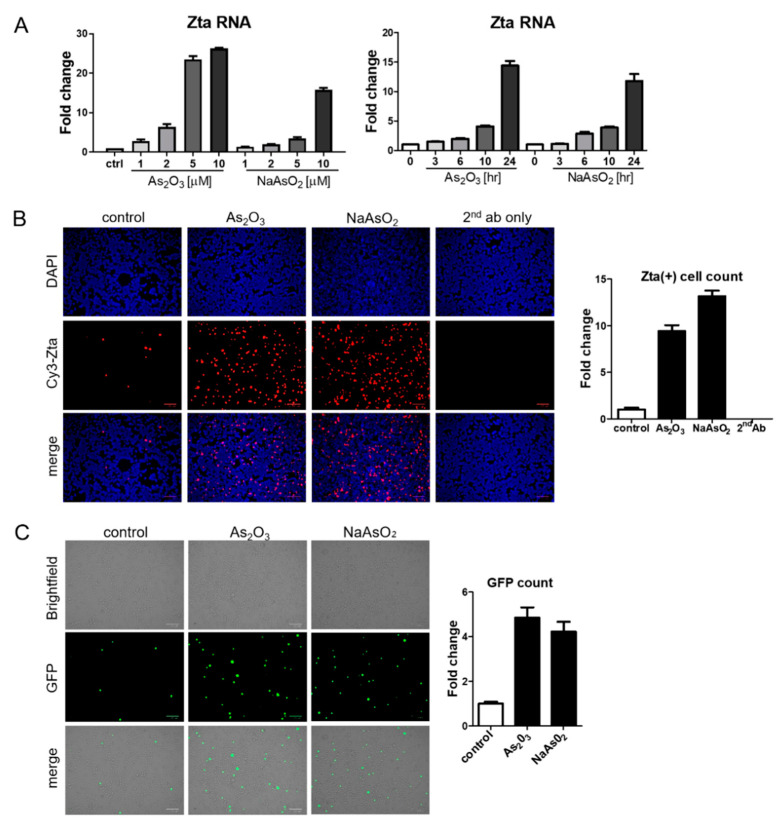
Arsenic induces EBV Zta expression in a lymphoma cell line. (**A**) BX1-Akata cells were treated with the indicated concentrations of As_2_O_3_ or NaAsO_2_ for 24 h, and isolated RNA was used for qRT-PCR with Zta primers (Left). BX1-Akata cells were treated with 10 µM As_2_O_3_ or NaAsO_2_ for the indicated time periods, and qRT-PCR was performed to detect Zta RNA levels (Right). (**B**,**C**) BX1-Akata cells were treated with 10 µM As_2_O_3_ or NaAsO_2_ for 24 h. Fluorescence was used to assess Zta (**B**) and GFP expression (**C**) following treatment. All scale bars on the bottom right represent 100 μm. Fold changes in numbers of positive cells are quantitated in the side panels.

**Figure 3 viruses-13-00812-f003:**
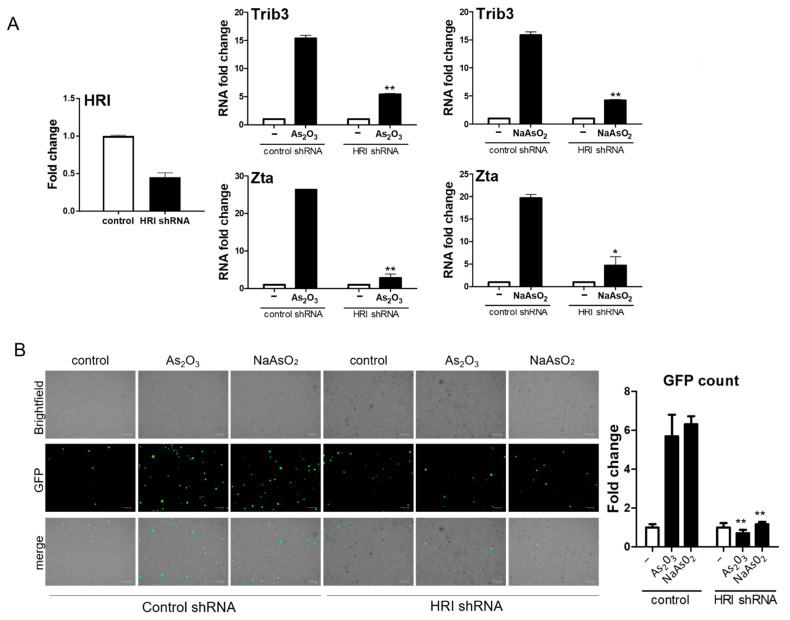
shRNA knockdown of HRI reduces EBV Zta activation by arsenic. BX1-Akata cells were transduced with shRNA lentiviral vectors designed to knockdown HRI gene expression or control shRNA lentiviral vectors. (**A**) HRI knockdown was confirmed by qRT-PCR. The rise in Trib3 and Zta RNA associated with arsenical treatment was blunted by HRI knockdown. (**B**) Fluorescence microscopy showed that HRI knockdown also eliminated any rise in GFP-positive cells after 24 h of arsenic treatment (* *p* < 0.05; ** *p* < 0.01).

**Figure 4 viruses-13-00812-f004:**
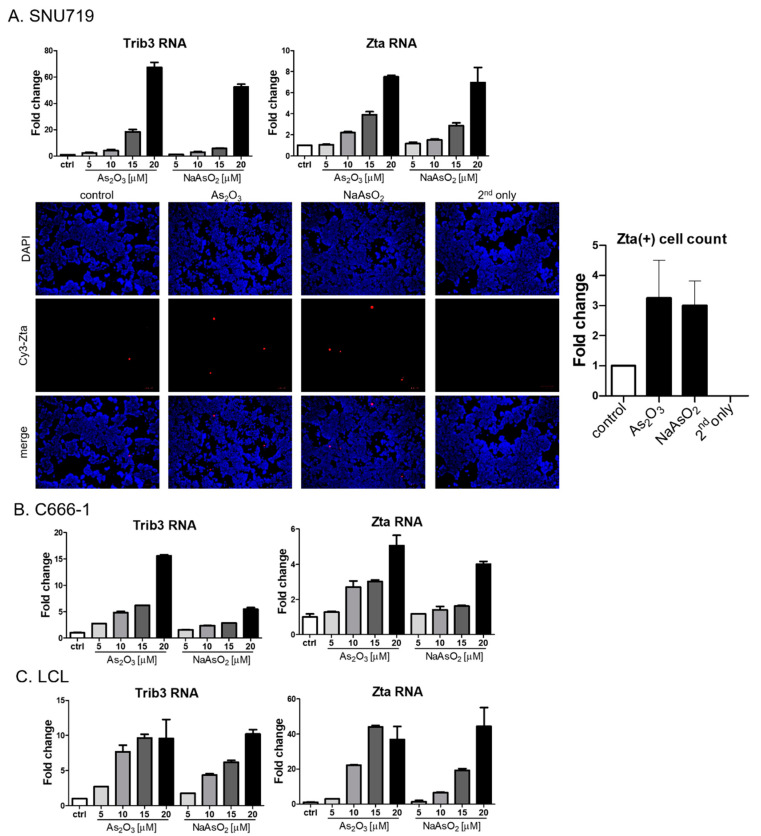
Arsenic activates EBV Zta expression in naturally infected epithelial cells and LCLs. (**A**) SNU719 cells were treated with indicated doses of arsenic for 24 h, and RNA was isolated. Trib3 (left) and Zta (right) RNA levels were determined by qRT-PCR. SNU719 cells were treated with 10 µM As_2_O_3_ or NaAsO_2_ for 24 h, and immunofluorescence was performed with an anti-Zta antibody (bottom). Zta-positive cells were quantified and graphed in the right. (**B**,**C**) C666-1 cells (**B**) and LCLs (**C**) were treated with indicated doses of arsenic for 24 h and isolated RNA was used for Trib3 (left) and Zta (right) RNA quantification.

**Figure 5 viruses-13-00812-f005:**
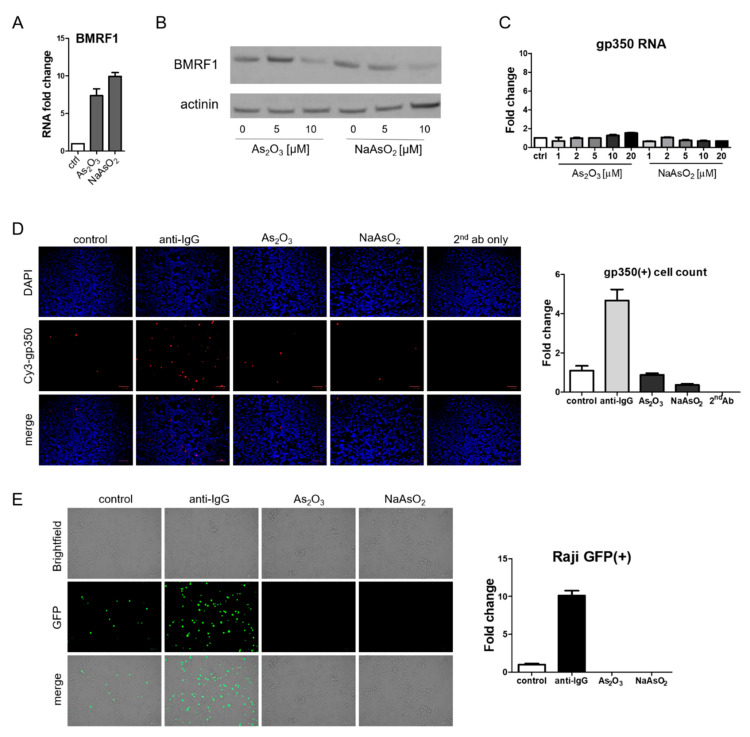
Arsenic does not induce late lytic gene expression and inhibits virion production. (**A**) BX1-Akata cells were treated with 10 µM As_2_O_3_ or NaAsO_2_ for 24 h, and BMRF1 RNA was measured by qRT-PCR. (**B**) BX1-Akata cells were treated with indicated doses of As_2_O_3_ or NaAsO_2_ for 24 h, and immunoblot was performed to detect BMRF1 protein expression. (**C**) BX1-Akata cells were treated with indicated doses of As_2_O_3_ or NaAsO_2_ for 24 h, and gp350 RNA was measured by qRT-PCR. (**D**) BX1-Akata cells were treated with either anti-IgG, 10 µM As_2_O_3_, or NaAsO_2_ for 24 h, and immunofluorescence was performed to detect gp350 protein level. (**E**) BX1-Akata cells were treated with either anti-IgG, 10 µM As_2_O_3_, or NaAsO_2_ for 4 days, and Raji cell infection assay was performed to determine infectious viral titers.

**Figure 6 viruses-13-00812-f006:**
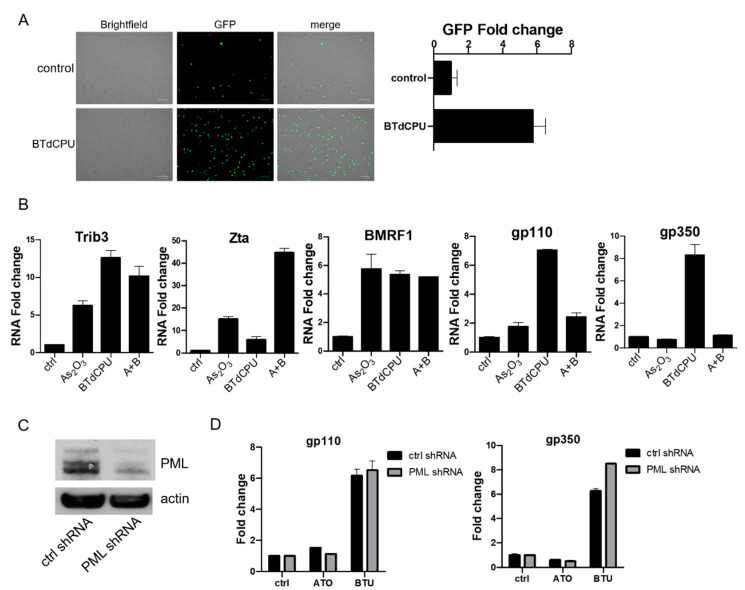
BTdCPU induces EBV lytic gene expression. (**A**) BX1-Aakta cells were treated with 10 µM BTdCPU, and fluorescence microscopy was used for detecting GFP positive cells after 24 h of treatment. (**B**) BX1-Akata cells were treated with either As_2_O_3_, BTdCPU, or in combination (**A**+**B**) for 24 h, and qRT-PCR was performed. (**C**,**D**) BX1-Akata cells were transduced with lentiviral particles designed to knockdown PML, and cells were selected by puromycin. Immunoblot was performed to detect PML expression (**C**), and qRT-PCR was performed 24 h after As_2_O_3_ or BTdCPU treatment (**D**).

## Data Availability

Data is contained within the article or Appendix A.

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
