# Peer review of "Arsenicals, the Integrated Stress Response, and Epstein–Barr Virus Lytic Gene Expression"

_viruses, 2021, doi:10.3390/v13050812_

Round 1

Reviewer 1 Report

This study investigated the impact of arsenic trioxide on the expression of EBV lytic genes. Treatment of arsenic trioxide in EBV-positive lymphoma cell lines, LCLs, and epithelial cancer cell lines cause the induction of the expression of BZLF1. However, arsenic treatment induced the expression of BMRF1 RNA, but not BMRF1 protein. The late gene, gp350, was not induced by the treatment and the production of EBV virions was also inhibited. The authors concluded that arsenic trioxide effects on EBV lytic cycle are mediated by other pathways.

General comments

            This paper primarily uses immunofluorescence to analyze protein expression quantitatively. The paper reaches a conclusion that both (proteins) EA-D and gP350/220 are not expressed. Immunofluorescence is not a good method to quantify protein expression; immunoblotting is. Therefore, this reviewer is not totally convinced that these early and late protein are not expressed. This paper should also include positive (western) controls, i.e. Akata treated and anti-IgG or TPA and sodium butyrate, to show that arsenic trioxide indeed does not induce the expression.

            It is difficult to understand as why the expression of Rta not studied. This can be important as transcription of BMRF1 and gp350 depends on Rta.

            It is well known that arsenic trioxide disperses PML nuclear bodies. More importantly, PML nuclear bodies are also known to influence SUMOylation. It is well known that SUMOylation influences the functions of EBV immediate-early proteins. The phenomena observed by the work can be explained by lack of protein posttranslational modification. This paper also ignores a fact that EBV capsid assembly requires PML-NBs. The lack of particle production is likely the absence of capsid assembly sites in the cells.

Specific comments

This study investigated how arsenic trioxide treatment influenced the expression of several EBV lytic genes. The results are confusing and difficult to interpret.

  1. Figure 2B shows the activation of Zta expression after arsenic trioxide treatment. The micrographs show that the percentage of the cells expressing Zta is high. The authors need to address this concern as lytic activation by TPA and sodium butyrate treatment does not seem to reach such a high activation rate.
  2. It is unclear as why GFP images are independently shown in Fig. 2C. The authors are supposed to show that Zta is only expressed in GFP-expressing cells. Showing GFP images separately does not serve this purpose.
  3. In Fig. 2, authors must show a western data to reveal the activation of Zta expression by arsenic trioxide.
  4. Lines 154: As indicated by in authors’ earlier paper, GFP is constitutively expressed in BX-Akata cells. If this is the case, why would the cells that express GFP increased after lytic activation?
  5. Fig. 3B: It is difficult to understand as why HRI knockdown decreases the number of BX-1 Akata cells.
  6. Fig. 5B, EA-D usually has a phosphorylated form which migrates a little slower than the non-phosphorylated EA-D. The authors must do two things here. A positive control “BX1-Akata treated with TPA and sodium butyrate” must be done to show phosphorylated EA-D. A whole blot instead of cropped images should be shown here.

Reviewer 2 Report

Induction of the lytic form of gene expression in latently infected Epstein-Barr virus (EBV) positive tumor cells has been proposed as a method for treating such tumors in humans.  Lytic reactivation is initiated by expression of the EBV immediate-early protein, BZLF1.  In this manuscript the authors show that As2O3 treatment of EBV-positive Burkitt tumors results in lytic EBV gene expression via its ability to activate the integrated stress response pathway, in particular through activation of the heme-regulated inhibitor (HRI) and eIF2α phosphorylation,  This effect is somewhat cell type dependent since it does not work as well in EBV transformed B cell lines (LCLs) or EBV-infected epithelial cells.  Interestingly, As203 treatment does not result in release of infectious viral particles in Burkitt cells (a potential therapeutic advantage). 
A small molecule HRI activator also led to increased lytic gene  expression in Burkitt cells but did did not block late lytic gene expression, suggesting that As2O3 effects on late lytic EBV gene expression are mediated through other pathways.

Overall the results of the paper are novel and interesting, and the experiments are well designed and interpreted.

Minor comment

It would be nice to include shRNA experiments showing whether specific transcription factors (such as ATF4 and/or XBP1) can be tied to the arsenic induction of BZLF1 expression in Burkitt cells

Reviewer 3 Report

This manuscript describes an interesting study analyzing the effects of As2O3 on EBV transcription and protein expression in several different cell lines. The study is well designed and the results presented clearly and logically.  One concern is the description of several of the methods.  This section must be written so that the readers could reproduce the experiments.  Unfortunately some of the methods are lacking in significant detail.  Specifically:

  1. qRT-PCR. How are the results normalized and the fold change calculated?
  2. Immunoblots: This section states that protein lysate was created from1x10E7 cells.  How many cell equivalents or amount of total protein was loaded into each lane?
  3. Raji infection assay: How many BX1-Akata cells were induced to produce virus and how much anti-IgG was added for the induction?  How much of the isolated virus was used to infect the Raji cells? How did the authors ensure that similar amounts of virus was added to each experiment?  
  4. What is the promoter driving the GFP in the BX1-Akata cells?
